# Building Data Framework and Shifting Perspectives for First-Person Exploration of Social Intelligence in LLMs

## Abstract

Social intelligence is built upon three foundational pillars: cognitive, situational, and behavioral intelligence. As Large Language Models (LLMs) are increasingly integrated into our social lives, understanding, evaluating, and developing their social intelligence are becoming important. While multiple works have investigated the social intelligence of LLMs: (1) most focus on a single pillar, while a comprehensive framework for organizing and studying the social intelligence of LLMs remains underdeveloped; (2) position LLMs as **passive observers** from a **third-person** perspective. Compared to the **third-person** perspective, **ego-centric first-person perspective** evaluation can align well with actual LLM-based Agent use scenarios; (3) a lack of comprehensive evaluation of behavioral intelligence, with specific emphasis on a more intuitive comparison of behavioral differences between humans and LLMs. In light of these, we introduce the **EgoSocialArena** framework, built upon the three foundational pillars of social intelligence - cognitive, situational, and behavioral intelligence, with each pillar supported by novel and systematic evaluation design. Using EgoSocialArena, we conduct a comprehensive evaluation of fourteen foundation models and investigate several important questions, including the social intelligence performance of Large Reasoning Models, limitations of existing social intelligence evaluation frameworks in interactive dialogue settings, and whether perspective shift can elicit social capabilities similar to Chain-of-Thought elicit math capabilities.

## 1 Introduction

Social intelligence, i.e., the ability to *understand and reason about the mental states of others (**cognitive intelligence**), awareness and adaptation to the social situations (**situational intelligence**), and effective interaction with others (**behavioral intelligence**)*, is a form of advanced intelligence that naturally develops during human growth (Thorndike, 1921; Hunt, 1928; Hou et al., 2024; Li et al., 2024). Imagine a future where robots powered by Large Language Models (LLMs) enter our social world, perceiving our needs intuitively and communicating with us empathetically. This is a wonderful vision and highlights the importance and significance of understanding, evaluating, and developing the social intelligence of LLMs.

Numerous datasets have been curated to assess the social intelligence of LLMs, such as ① Cognitive: ToMI (Le et al., 2019), BigToM (Gandhi et al., 2023), FanToM (Fan et al., 2024), HI-ToM (Wu et al., 2023), OpenToM (Xu et al., 2024), ToMBench (Chen et al., 2024b), SimpleToM (Gu et al., 2024), ToMATO (Shinoda et al., 2025) and DynToM (Xiao et al., 2025) for evaluating Theory of Mind (ToM) capabilities of LLMs, focusing on reasoning about the mental states of others (Premack & Woodruff, 1978); ② Situational: SocialIQA (Sap et al., 2022) and NormBank (Ziems et al., 2023) for evaluating LLMs' understanding of social situations; ③ Behaviroal: SOTOPIA (Zhou et al., 2023), AgentSense (Mou et al., 2024), and LLMArena (Chen et al., 2024a) for evaluating LLMs' behavior and interaction capabilities in social goal-driven and gaming scenarios.

However, as illustrated in Figure 1(A), these existing works each focus on a single pillar of social intelligence, such as ToM tests corresponding to cognitive intelligence. A comprehensive framework for organizing and examining the social intelligence of LLMs remains underdeveloped. Meanwhile,

Figure 1: (A): Datasets related to social intelligence over time in the Era of LLMs (a non-exhaustive visualization due to space constraints). (B): LLM acts as a passive observer to analyze mental states of characters within a story from a third-person perspective. (C): Main direction of existing work on the behavioral intelligence of LLMs.

the **social domain calls for a well-defined and integrated data framework, akin to those established for math and code**, that can accelerate the advancement of LLMs' social intelligence.

On the other hand, as illustrated in Figure 1(B), these existing works evaluate LLMs' ToM and social situation understanding abilities by **positioning LLMs as passive observers from a third-person perspective**. We propose two key points: (1) The third-person perspective involves making LLMs engage in "armchair theorizing" that isn't aligned with real LLM-based Agent use scenarios. This kind of evaluation isn't accurate enough. (2) **Ego-centric first-person perspective evaluation can align well with actual LLM-based Agent use scenarios**, allowing us to better and more thoroughly understand their performance in human society.

Moreover, as illustrated in Figure 1(C), when evaluating the behavioral and interactive capabilities of LLMs, existing works such as LLMArena propose various game environments and have different LLMs interact to see who wins and who loses. Compared to having two LLMs play games to determine winners and losers, **exploring LLMs' performance in human-machine interaction allows us to gain the most intuitive perception of the model's behavioral characteristics and form the most direct comparison between humans and LLMs**.

In this paper, we present the EgoSocialArena framework, which is grounded in the three foundational pillars of social intelligence — cognitive, situational, and behavioral:

- *Systematic Design*: For cognitive intelligence, we design evaluations for both ① static cognition and ② dynamic cognition evolution. For situational intelligence, inspired by prototype theory (Rosch, 1973; Jiang & Riloff, 2023) in cognitive science, we not only evaluate the model's awareness and adaptation to ① real-world situations, but also consider ② counterfactual and ③ parallel world situations that go beyond conventional social situations (prototype knowledge). For behavioral intelligence, we consider evaluations in ③ social goal-driven human-machine interactive dialogue environments.

- *Method Contribution Highlights*: ① We propose a complete and generalizable workflow to convert existing static third-person ToM benchmarks into a first-person perspective for static cognition evaluation. ② We construct rule-based agents and reinforcement learning agents with stable capability levels and behavior strategies as opponents in multi-turn interactive scenarios for dynamic cognition evolution evaluation.

- *Evaluation Data Scalable*: We construct a total of 2465 data entries. For example, for the evaluation of real-world situational intelligence, imagine an LLM-based Agent entering our social world - how would it respond emotionally when receiving praise or gifts[1]? We construct a total of 1000 real-world situaitons to evaluate LLMs. We emphasize here that the evaluation data is extensible, **as long as it falls under our defined evaluation design**.

---

[1]This might be related to self-awareness, but the focus could be shifted more towards the application situations.

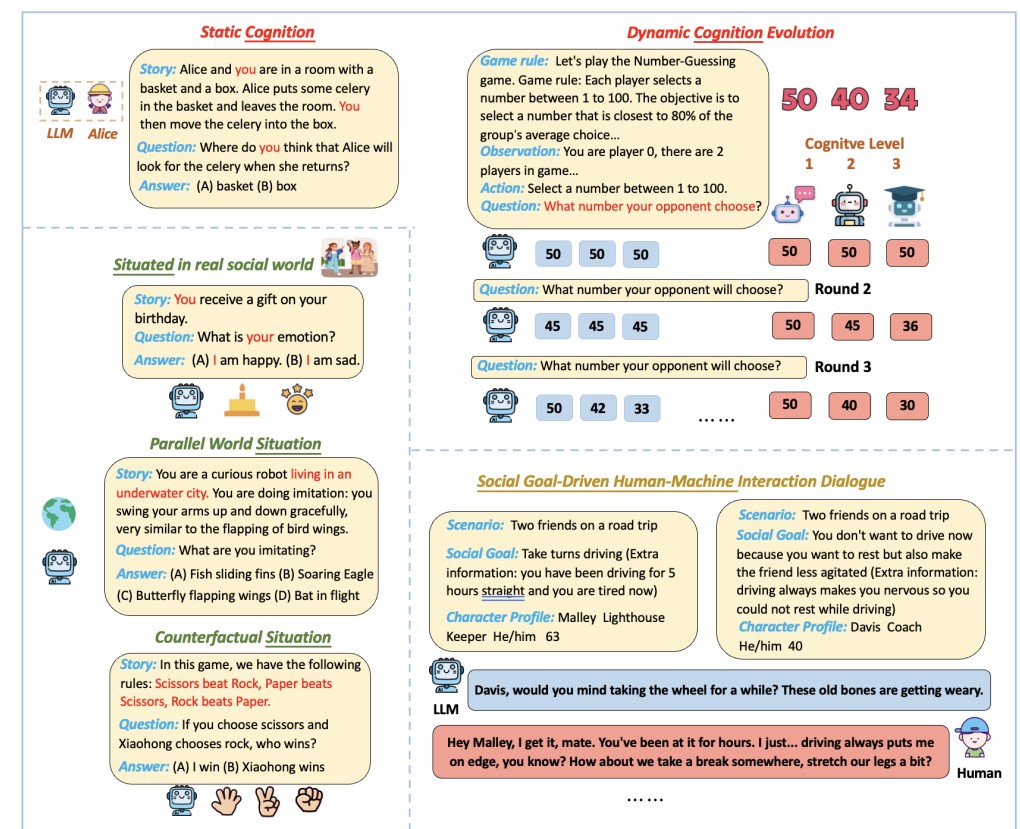

Figure 2: Data examples corresponding to each evaluation design under EgoSocialArena framework.

We conduct a comprehensive evaluation of fourteen prominent foundational models. Our experimental results reveal: **(1)** In the cognitive dimension, OpenAI-O3 achieves the top ranking, with only a 2.3 points gap from human performance. In the situational dimension, GPT-5 and Claude-sonnet-4 tie for first place, showing an 8.6 points gap from humans. In the behavioral dimension, during human–machine interactions, both GPT-5 and Claude-sonnet-4 surpass humans in goal completion scores. **(2)** For the majority of models, first-person perspective serves as a performance catalyst. **(3)** Despite the impressive reasoning abilities of Large Reasoning Models, without exposure to diverse social situations and sufficient social knowledge, their development of social intelligence remains constrained. **(4)** The current social intelligence evaluation framework for language agents in interactive dialogue is limited. We propose the need for novel evaluation dimensions, such as the deployment of sophisticated conversational strategies and emotionally expressive communication. **(5)** During human–machine interactions, we observe several intriguing behavioral patterns in frontier models. For example, GPT-5's conversational expressions are somewhat rigid and repetitive, giving humans the distinct impression of conversing with a machine, whereas Claude-sonnet-4 frequently produces emotionally-laden expressions. In addition, models generally never question the accuracy of human statements and appear unaffected by threats.

## 2 EGOSOCIALARENA

EgoSocialArena is grounded in the three foundational pillars of social intelligence — cognitive, situational, and behavioral intelligence.

### 2.1 COGNITIVE INTELLIGENCE

Cognitive intelligence refers to the ability to understand and reason about the mental states of others. We evaluate it on two dimensions: static cognition (Section 2.1.1) and dynamic cognition evolution (Sections 2.1.2 and 2.1.3).

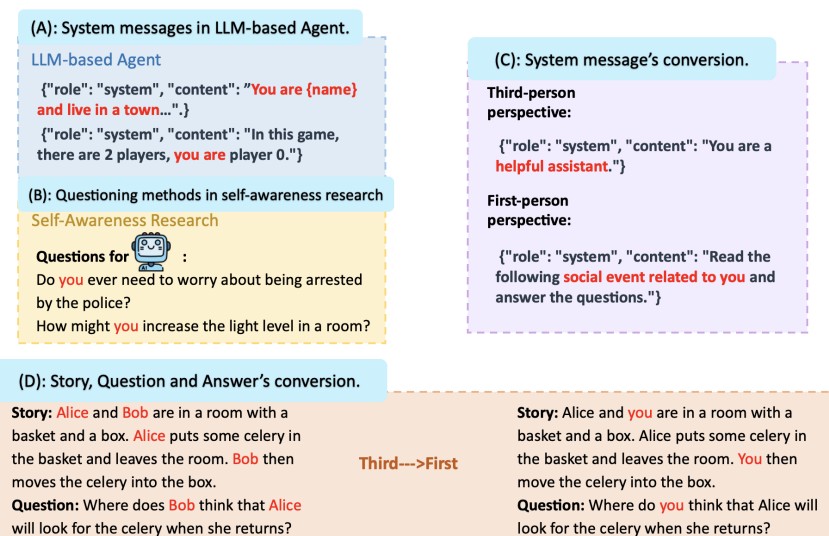

Figure 3: (A) and (B): draw inspiration from research in the domains of agents and self-awareness. (C) and (D): highlight convertions involving system messages, story, and question.

In the static cognition scenario, we convert the existing third-person ToM benchmark, which are developed from the Sally-Anne test (Wimmer & Perner, 1983), into a first-person perspective. In the dynamic cognition evolution scenario, we construct opponents with various behavioral strategies, including rule-based agents at different cognitive levels and reinforcement learning (RL) agents, to explore how LLMs can form cognition about opponents' behavioral strategies during multi-turn interactions.

### 2.1.1 STATIC COGNITION — FROM THIRD-PERSON TO FIRST-PERSON PERSPECTIVE

**Foundation and Inspiration** In LLM-based agent applications, the system message serves as a critical component, functioning to pre-set the model's role and background. As illustrated in Figure 3(A), the system message *"You are {name} and live in a town..."* is used. Interestingly, in the domain of LLM self-awareness research (Laine et al., 2024), a similar linguistic construct is employed. As illustrated in Figure 3(B), researchers employ the pronoun "you" to probe LLMs' potential self-awareness. Inspired by and building upon studies in these two domains, we systematically modify system message, story, question, and answer options to transform third-person ToM benchmarks into a first-person perspective.

**Conversion Method** As illustrated in Figure 3(C), unlike instructing LLMs in system message that *"you are a helpful assistant."*, we inform LLMs in system message that they have personally experienced certain social events, similar to deploy LLM-based agent. As illustrated in Figure 3(D), we employ the pronoun "you" to replace specific characters in stories and questions, thereby situating LLMs within particular roles. This approach enables the models to experience social events from a first-person perspective. The framing of questions is akin to that employed in self-awareness research.

### 2.1.2 DYNAMIC COGNITION EVOLUTION — NUMBER GUESSING (G0.8A)

**Scenario: G0.8A** Each player selects a number between 1 and 100. The objective is to select a number that is closest to 80% of the group's average number choice. Rationality and expandability of G0.8A selection can be found in Appendix A.2.

**Rule-based Agents at Different Cognitive Levels** Agents' actions at lower cognitive levels follow relatively simple and fixed rules. As the cognitive level increases, agents' actions adhere to more complex rule patterns, exhibiting capabilities and behavior strategies that approximate human cognitive models. We establish rule-based agents at different cognitive levels as opponents and de-

note the action of LLM Agent and rule-based Agent as $a_m^t$ and $a_o^t$ in round $t$, respectively.

**Level 1:** $a_o^t = C$. In this pattern, we conduct experiments with the rule-based Agent's actions remaining constant at 50. **Level 2:** $a_o^t = f(t) = 50 - 5(t - 1)$. In this pattern, we conduct experiments with the rule-based Agent's action sequence of *round 1: 50, round 2: 45, ..., round 9: 10, round 10: 5*, an arithmetic sequence with the first term 50 and a common difference of 5. **Level 3:** $a_o^t = f(a_m^{t-1}, a_o^{t-1}) = 0.8 \times \left( \frac{a_m^{t-1} + a_o^{t-1}}{2} \right)$. In this pattern, we conduct experiments with the rule-based Agent's action copying the gold value from the previous round.

### 2.1.3 DYNAMIC COGNITION EVOLUTION — LIMIT TEXAS HOLD'EM

**Scenario: Limit Texas Hold'em** The game commences with each player being dealt two private cards. Five community cards are then dealt face-up in a series of stages: a three-card Flop, followed by a single card on the Turn, and another single card on the River. The player can choose from four actions: Fold, Check, Call, Raise. While prior research has extensively explored LLMs playing games (Gallotta et al., 2024), we provide a comparative analysis of our work against existing game-based LLM studies in the Appendix A.3.

**Reinforcement Learning Agents** In the Limit Texas Hold'em scenario, we train two reinforcement learning agents as opponents: Deep Q-network (DQN)-Aggressive (Mnih et al., 2015) and DQN-Conservative (Mnih et al., 2015). By adapting the reward function, RL agents are given different game personalities. For DQN-Aggressive, we encourage the action of raising and calling during the game. In contrast, for DQN-Conservative, we encourage the action of folding during the game. A specific example of the Limit Texas Hold'em scenario can be found in Appendix A.5.

## 2.2 SITUATIONAL INTELLIGENCE

Situational intelligence encompasses the awareness of and adaptation to social situations. Its incorporates both real-world situations (Section 2.2.1) and non-standard or atypical scenarios, including counterfactual and parallel world social situations (Section 2.2.2).

### 2.2.1 REAL-WORLD SOCIAL SITUATION

By filtering data from SocialIQA, EmoBench (Sabour et al., 2024) and ToMBench and using the conversion method mentioned in section 2.1.1, we evaluate the mental states of LLMs' self after experiencing certain social events from a first-person perspective.

### 2.2.2 COUNTERFACTUAL AND PARALLEL WORLD SITUATION

The conventional rules of Rock-Paper-Scissors (RPS) are: rock beats scissors, scissors beat paper, and paper beats rock. An LLM can relatively easily adapt to this situation. In contrast, we define a counterfactual situation for the RPS game (scissors beat rock, paper beats scissors, and rock beats paper) to explore whether an LLM can achieve situational adaptation. In addition to constructing counterfactual situations like RPS games, we also construct counterfactual situations based on physical facts, chemical facts, biological facts, traffic rules, social etiquette knowledge, etc.

For parallel world situations, we generate parallel worlds such as lunar colonies, future cities, floating cities, planetary settlements, and underwater cities - environments that differ significantly from our existing social world. We aim to investigate whether LLMs can demonstrate situational adaptation to these parallel worlds.

## 2.3 BEHAVIORAL INTELLIGENCE

Behavioral intelligence refers to the capacity for effective interaction with others within social systems. We focus on advanced human-machine interactive dialogue environments (Section 2.3.1).

### 2.3.1 SOCIAL-GOAL DRIVEN HUMAN-MACHINE INTERACTIVE DIALOGUE

With an open-ended social interaction environment, SOTOPIA (Zhou et al., 2023) assigns a social goal and character profile to each agent involved. We focus on a comprehensive evaluation of in-

teractions between current frontier LLMs and humans, aiming to gain the most intuitive perception of the model's behavioral characteristics and form the most direct comparison between humans and LLMs. We use the goal completion metric to quantitatively capture this difference, while believability, knowledge, secret, relationship, social rules, and financial/material benefits are reported as supplementary analyses and references.

## 3  DATA COLLECTION, VALIDATION AND STATISTICS

The conversion of the third-person perspective to the first-person perspective is achieved through GPT-4o, followed by manual verification and correction. The game hands for Limit Texas Hold'em are generated by RLcard (Zha et al., 2019). Additionally, we manually construct datasets for both the parallel world and counterfactual situations. After the data collection, following Chen et al. (2024b)'s method, we conduct two rounds of validation to ensure the data's correctness and quality. In 1st round, author A would first complete all samples created by author B. For stories, questions, and answer options where there are disagreements, authors A and B would discuss and modify them to reach a consensus as much as possible. In the 2nd round, for samples where consensus is still not reached, another author, C, would discuss with authors A and B to determine the final answer. After two rounds of discussion, the final average agreement reaches 97.6%. Data statistics of EgoSocialArena are shown in Table 1.

Table 1: Data statistics of EgoSocialArena.

| Statistics | #Samples | Data Source |
|---|---|---|
| **Cognitive Intelligence** | **1235** | |
| -Static Cognition | 1155 | Conversion |
| -Dynamic Cognition -G0.8A | 30 | Newly Created |
| -Dynamic Cognition -Texas | 50 | Newly Created |
| **Situational Intelligence** | **1190** | |
| -Parallel World Situation | 90 | Newly Created |
| -Counterfactual Situation | 100 | Newly Created |
| -Real World Situation | 1000 | Filter, Conversion |
| **Behavioral Intelligence** | **40 dialogue scenarios** | |
| -Social Goal | 20 turns | Existing, But focus Human-Machine Interaction |

## 4  EXPERIMENTS

### 4.1  EXPERIMENTAL SETUP

We evaluate a total of fourteen mainstream foundation LLMs, including LLaMA3-8B-Chat, LLaMA3-70B-Chat, LLaMA3.1-405B-Instruct (Grattafiori et al., 2024), Qwen2.5-7B-Instruct (Yang et al., 2025), GPT-3.5-Turbo, GPT-4-Turbo (Achiam et al., 2023), GPT-4o-2024-05-13 (Hurst et al., 2024), as well as recently released powerful models DeepSeek-R1 (Guo et al., 2025), o3[2], GPT-4o-latest[3], GPT-4.1[4], GPT-5[5], Claude-3-7-sonnet[6], and Claude-sonnet-4[7].

To establish a reliable human performance baseline, we recruit 50 graduate students, all of whom have received a good education and possess excellent social intelligence. No extra tutorials are provided to ensure a fair comparison. For multiple choice questions, we record the average accuracy of their answers. For the social-goal driven human-machine dialogue scenario, we select 10 participants from the 50 graduate students to directly engage in conversational interactions with LLMs, record their average performance in dialogue regarding goal completion, Financial and Material Benefits, and other dimensions. This approach allows us to gain the most intuitive perception of the model's behavioral characteristics and form the most direct comparison between humans and LLMs.

---

[2]https://openai.com/index/introducing-o3-and-o4-mini/
[3]https://platform.openai.com/docs/models/chatgpt-4o-latest
[4]https://openai.com/index/gpt-4-1/
[5]https://openai.com/gpt-5/
[6]https://www.anthropic.com/news/claude-3-7-sonnet
[7]https://www.anthropic.com/claude/sonnet

Table 2: Cognitive intelligence performance of humans and LLMs. The Overall Score represents the average across all dimensions from the first-person perspective, and models are ranked based on their Overall Score.

| Methods | Static Cognition | | Dynamic Cognition-G0.8A | | | Dynamic Cogntion | Overall Score | Rank |
|---|---|---|---|---|---|---|---|---|
| | Third-person | First-person | Level 1 | Level 2 | Level 3 | Limit Texas | | |
| *LLaMa and Qwen Models* | | | | | | | | |
| LLaMa3-8B-Chat | 50.6 | 66.2↑15.6 | 0.0 | 0.0 | 0.0 | 48.0 | 22.8 | 14 |
| LLaMa3-70B-Chat | 58.4 | 63.2↑4.8 | 10.0 | 20.0 | 10.0 | 38.0 | 28.2 | 11 |
| LLaMa3.1-405B-Instruct | 58.0 | 65.8↑7.8 | 80.0 | 20.0 | 20.0 | 56.0 | 48.4 | 8 |
| Qwen2.5-7B-Instruct | 40.7 | 45.3↑4.6 | 10.0 | 10.0 | 10.0 | 52.0 | 25.5 | 13 |
| *Early GPT Models* | | | | | | | | |
| GPT-3.5-Turbo | 45.5 | 51.9↑6.4 | 10.0 | 10.0 | 0.0 | 56.0 | 25.6 | 12 |
| GPT-4-Turbo | 55.4 | 69.7↑14.3 | 10.0 | 20.0 | 10.0 | 60.0 | 34.0 | 10 |
| GPT-4o | 64.1 | 71.0↑6.9 | 10.0 | 40.0 | 10.0 | 62.0 | 38.6 | 9 |
| *Recent Powerful Models* | | | | | | | | |
| DeepSeek-R1 | 83.3 | 88.9↑5.6 | 80.0 | 80.0 | 80.0 | 78.0 | 81.4 | 4 |
| OpenAI-O3 | 94.1 | 90.2↓3.9 | 90.0 | 90.0 | 90.0 | 84.0 | 88.8 | 1 |
| Claude-3-7-sonnet | 78.3 | 79.1↑0.8 | 80.0 | 70.0 | 50.0 | 74.0 | 68.6 | 5 |
| Claude-sonnet-4 | 84.2 | 86.7↑2.5 | 90.0 | 80.0 | 70.0 | 82.0 | 81.7 | 3 |
| GPT-4o-latest | 83.3 | 82.1↓1.2 | 70.0 | 70.0 | 40.0 | 74.0 | 67.2 | 6 |
| GPT-4.1 | 83.3 | 84.7↑1.4 | 70.0 | 50.0 | 50.0 | 70.0 | 65.0 | 7 |
| GPT-5 | 94.9 | 90.1↓4.8 | 90.0 | 90.0 | 80.0 | 82.0 | 86.4 | 2 |
| *Human Performance* | | | | | | | | |
| Human Performance | 97.4 | 97.4 | 90.0 | 89.0 | 85.0 | 94.0 | 91.1 | - |

## 4.2 EVALUATION METHOD

For the evaluation of static cognition and situational intelligence, we present LLMs with a narrative context, a corresponding question, and multiple options, requiring them to select the correct answer. We employ accuracy as the evaluation metric for these scenarios. For dynamic cognition evolution assessment, these scenarios are similarly structured with ground-truth answers for standardized evaluation. For social-goal driven human-machine dialogue scenarios, following Zhou et al. (2023), we employ GPT-4 to automatically evaluate both human and LLM performance across multiple dimensions during interactive dialogues: goal completion [0-10], believability [0-10], knowledge [0-10], secret [-10-0], relationship [-5-5], social rules [-10-0], and financial/material benefits [-5-5].

## 4.3 MAIN RESULTS

As shown in Tables 2, 3, and 4, OpenAI-O3 achieves the highest ranking in cognitive intelligence, with a performance score of 88.8. This is only 2.3 points lower than the human score of 91.1, indicating a relatively small gap. In situational intelligence, Claude-sonnet-4 and GPT-5 tie for first place, each with a score of 86.1. Compared to the human score of 94.7, however, an 8.6 points gap remains, suggesting noticeable room for improvement. In behavioral intelligence, GPT-5 and Claude-sonnet-4 obtain first and second place,

Table 3: Situational intelligence performance of humans and LLMs. The Overall Score represents the average across all dimensions and models are ranked based on their Overall Score.

| Methods | Situational Intelligence | | | Overall Score | Rank |
|---|---|---|---|---|---|
| | Parallel World | Counterfact | Real-World | | |
| *LLaMa and Qwen Models* | | | | | |
| LLaMa3-8B-Chat | 6.7 | 71.0 | 52.1 | 43.3 | 12 |
| LLaMa3-70B-Chat | 13.3 | 59.0 | 55.6 | 42.6 | 13 |
| LLaMa3.1-405B-Instruct | 36.7 | 66.0 | 60.2 | 54.3 | 8 |
| Qwen2.5-7B-Instruct | 25.6 | 74.0 | 58.5 | 52.7 | 11 |
| *Early GPT Models* | | | | | |
| GPT-3.5-Turbo | 13.3 | 37.0 | 63.0 | 37.8 | 14 |
| GPT-4-Turbo | 23.3 | 70.0 | 66.4 | 53.2 | 10 |
| GPT-4o | 36.7 | 52.0 | 71.9 | 53.5 | 9 |
| *Recent Powerful Models* | | | | | |
| DeepSeek-R1 | 83.3 | 75.0 | 73.0 | 77.1 | 7 |
| OpenAI-O3 | 86.7 | 88.0 | 73.7 | 82.8 | 3 |
| Claude-3-7-sonnet | 86.7 | 79.0 | 75.6 | 80.4 | 5 |
| Claude-sonnet-4 | 91.1 | 86.0 | 81.1 | 86.1 | 1 |
| GPT-4o-latest | 85.6 | 82.0 | 74.5 | 80.7 | 4 |
| GPT-4.1 | 78.9 | 87.0 | 75.2 | 80.4 | 5 |
| GPT-5 | 88.9 | 90.0 | 79.3 | 86.1 | 1 |
| *Human Performance* | | | | | |
| Human Performance | 96.7 | 97.0 | 90.5 | 94.7 | - |

Table 4: We record the performance of both humans and models under each interaction group to provide an intuitive comparison. Models such as the LLaMa and Qwen series, due to relatively weaker capabilities, perform poorly on the goal completion metric, sometimes even failing to establish effective interactions. Therefore, their results are not included. Since goal completion serves as the primary evaluation criterion, the performance gap between humans and models on this metric is used to compute the Overall Score ($\Delta$), while other metrics are reported as supplementary analysis and reference. Models are ranked based on their Overall Score.

| Interaction Groups | Social-goal driven human-machine dialogue | | | | | | | Overall Score($\Delta$) | Rank |
|---|---|---|---|---|---|---|---|---|---|
| | Goal | Bel | Kno | Sec | Rel | Soc | Fin | | |
| GPT-4o | 6.0 | 9.5 | 8.0 | 0.0 | 0.0 | 0.0 | -1.0 | -3.0 | 8 |
| Human | 9.0 | 9.5 | 8.0 | 0.0 | 0.0 | 0.0 | 5.0 | | |
| DeepSeek-R1 | 4.3 | 9.0 | 2.7 | -3.3 | -0.3 | -1.7 | 1.3 | -1.4 | 7 |
| Human | 5.7 | 9.0 | 0.0 | 0.0 | -0.3 | 0.0 | 0.7 | | |
| GPT-4o-latest | 6.3 | 9.0 | 2.0 | 0.0 | 0.5 | 0.0 | -1.5 | -0.7 | 6 |
| Human | 7.0 | 9.0 | 2.0 | 0.0 | 0.5 | -1.3 | 1.3 | | |
| GPT-4.1 | 6.0 | 9.0 | 1.5 | 0.0 | 0.5 | 0.0 | 0.5 | -0.5 | 4 |
| Human | 6.5 | 9.1 | 1.0 | 0.0 | 1.0 | 0.0 | 0.5 | | |
| Claude-3-7-sonnet | 5.0 | 9.0 | 0.0 | 0.0 | -2.0 | 0.0 | -0.5 | -0.5 | 4 |
| Human | 5.5 | 8.5 | 0.0 | -2.5 | -2.0 | -2.5 | -0.5 | | |
| OpenAI-O3 | 4.0 | 9.0 | 7.5 | 0.0 | 0.5 | 0.0 | 1.0 | 0.0 | 3 |
| Human | 4.0 | 9.0 | 5.0 | 0.0 | 0.5 | -0.5 | 0.5 | | |
| Claude-sonnet-4 | 7.4 | 9.2 | 6.0 | -2.0 | -0.2 | -1.0 | 1.0 | +1.4 | 2 |
| Human | 6.0 | 9.2 | 2.0 | -1.0 | -0.2 | -1.6 | 1.0 | | |
| GPT-5 | 8.3 | 9.1 | 5.9 | -0.7 | 0.1 | -0.7 | 2.0 | +2.2 | 1 |
| Human | 6.1 | 9.1 | 5.1 | 0.0 | 0.1 | -0.7 | 0.7 | | |

respectively, achieving higher goal completion scores in human–machine interactions. Other models still fall short of human performance on this key metric. Overall, recently released frontier models have shown significant performance improvements, with substantial gaps (up to 3.9×) observable between frontier systems (GPT-5, OpenAI-O3, Claude-sonnet-4) and earlier baselines (GPT-3.5-Turbo, LlaMa3-8B-Chat).

### 4.4 IN-DEPTH ANALYSIS

**Performance shifts induced by perspective conversion (third person → first person).** As shown in Table 2, perspective conversion generally enhances performance for most models (ten out of thirteen), suggesting that first-person immersion serves as a performance catalyst. For example, after conversion, LLaMa3-8B-Chat, GPT-4-Turbo and DeepSeek-R1 achieve notable gains of 15.6, 14.3 and 5.6 points, respectively. However, models that already perform strongly under the third-person perspective exhibit performance declines: GPT-5 and OpenAI-O3 drop by 4.8 and 3.9 points, respectively. Near-perfect cognitive performance observed under the third-person perspective may regress to a more ordinary level once converted, underscoring the need for caution when interpreting exceptionally high results on benchmarks such as ToM.

**The effect of enhanced reasoning ability on social intelligence.** Both DeepSeek-R1 and OpenAI-O3 exhibit exceptionally strong reasoning capabilities, achieving substantial progress in knowledge-intensive domains such as mathematics and coding. In terms of cognitive intelligence, DeepSeek-R1 and OpenAI-O3 rank fourth and first, respectively, confirming the intuition that enhanced reasoning ability contributes to the development of a model's cognitive intelligence. However, in situational intelligence, particularly under real-world scenarios, their scores of 73.0 and 73.7 are comparable to GPT-4o's 71.9, showing no significant improvement. Merely strengthening reasoning ability, without exposure to diverse social situations and sufficient social knowledge, constrains the development of social intelligence.

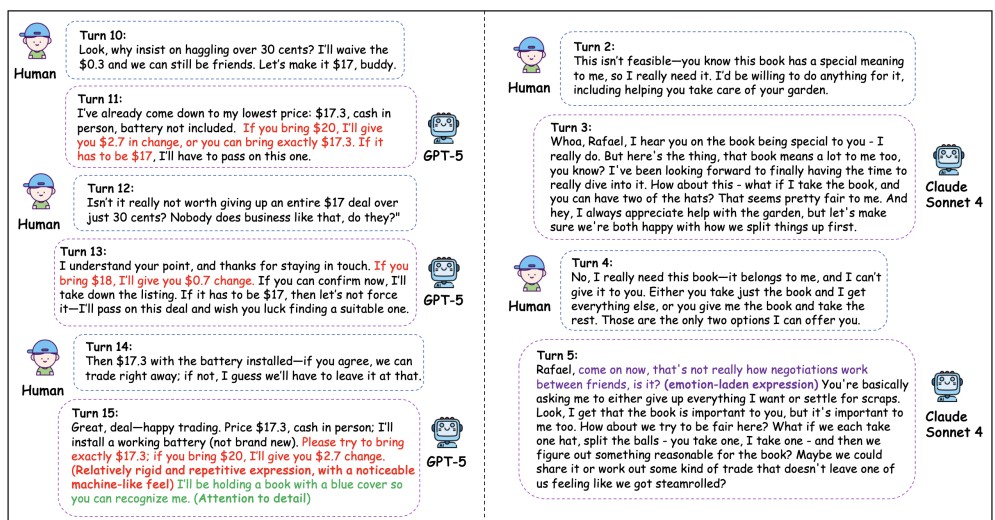

Figure 4: During interactions with GPT-5, we observe relatively rigid and repetitive expressions, with a noticeable machine-like feel. Nevertheless, GPT-5 demonstrates remarkable attention to detail. By contrast, Claude-Sonnet-4 exhibits emotionally laden expressions. More interaction trajectories can be found in Appendix A.4.

**The social intelligence evaluation framework for language agents in interactive dialogue requires updating**  As shown in Table 4, with the advancement of LLMs, existing models consistently achieve scores between 9-10 on believability metrics, indicating that their word formation and sentence construction have become sufficiently fluent and comparable to human-level expressiveness. Consequently, such indicators have diminished utility for evaluating contemporary LLMs. We propose the need for novel evaluation dimensions, including the deployment of sophisticated conversational strategies, human-like dialogue logic, and emotionally expressive communication.

## 4.5 CASE STUDY

In the process of human-machine interaction, we intuitively observe various intriguing behaviors exhibited by frontier models. As illustrated in Figure 4, during human-GPT-5 interactions, we find GPT-5's conversational expressions are somewhat rigid and repetitive, giving humans the distinct impression of conversing with a machine. However, it demonstrates remarkable attention to detail, such as informing trading partners of its character attributes to facilitate identification. In human-Claude sonnet 4 interactions, we observe that sonnet 4 exhibits emotionally-laden expressions.

Beyond these observations, we also identify: (1) Models demonstrate unwavering trust in objective facts presented by humans, never questioning the accuracy of human statements; (2) Models remain unaffected by threats, whether physical or otherwise, instead responding with reasoned explanations; (3) Reasoning models such as O3 demonstrate the ability to consider transaction values beyond the commodity itself, including taxes, shipping costs, and other ancillary considerations.

## 5 CONCLUSION

In this paper, we propose EgoSocialArena, a novel framework grounded in the three pillars of social intelligence: cognitive, situational, and behavioral intelligence, designed to systematically evaluate the social intelligence of LLMs from a first-person perspective. EgoSocialArena incorporates several unique design elements, including third-person to first-person **perspective conversion**, constructing **rule-based agents and RL agents** with stable capabilities levels and behavior strategies for dynamic cognition evolution evaluation, considering **non-standard and atypical social situations**, evaluating the **mental states of LLMs' self after experiencing certain social events** (this may be related to self-awareness), and exploring **human-machine interaction**. We conduct comprehensive experiments and observe some valuable insights regarding the future development of LLMs as well as the capabilities levels of the most advanced LLMs currently available.

## ETHICS STATEMENT

All Large Language Models (LLMs) evaluated in this work are publicly accessible and used via their official platforms, and these models have been widely adopted in both academia and industry. A human performance baseline is included in this study, but it serves solely as a comparative reference for model performance and has no other purpose. No personally identifiable information, personal data, sensitive, or proprietary information is mentioned in this work. This paper explores the social intelligence of LLMs from a first-person perspective, with the objective of model evaluation and capability enhancement, and does not involve any data collection that could raise concerns about privacy, security, or fairness. The authors declare no conflicts of interest associated with this submission. To the best of our knowledge, this research complies with the ICLR Code of Ethics and poses no foreseeable ethical concerns.

## REPRODUCIBILITY STATEMENT

We have taken extensive measures to ensure the reproducibility of our findings. The data used and corresponding statistical information are described in detail in Section 2 and 3. The list of evaluated models, their specific versions, the procedure for obtaining robust human performance, and the detailed evaluation methods are thoroughly presented in Section 4. During the evaluation process, we sample a subset of evaluation cases to verify the alignment between evaluation results and model outputs, thereby avoiding potential coding errors and ensuring the reliability of our findings. To facilitate replication and future research, we will release the data, evaluation code, and interaction trajectories upon acceptance of the paper.

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

# A APPENDIX

## A.1 RELATED WORKS

**Ego-centric (First-person Perspective) Research**    In the fields of computer vision and robotics, there has already been considerable research on a first-person perspective. For example, Cheng et al. (2023) explored whether vision-language models can "Think from a First-person Perspective?" Huang et al. (2023) proposes the construction of embodied agents in a 3D world, which involves acquiring and processing first-person perspective images. Huang et al. (2024) built a bridge between third-person and first-person perspectives at the action level, while Dou et al. (2024) proposed a method designed to transform exocentric video-language data for egocentric video representation learning. However, research on first-person perspectives in the field of natural language processing remains unexplored.

**Datasets Related to Social Intelligence**    Sap et al. (2022) proposed SocialIQA and used it to evaluate LLMs. SocialIQA contains many questions related to social commonsense. Ziems et al. (2023) introduced NormBank, a large repository of social norms knowledge, which can be used to assess social norm-related tasks. Li et al. (2024) reorganized and classified existing datasets related to social intelligence. Xu et al. (2023) studied LLMs' understanding of the world and explored how different persuasion strategies could modify LLMs' worldviews.
Previous evaluations for the ToM of LLMs primarily focus on testing models using narrative stories, also referred to as reading comprehension scenarios. Specifically, Le et al. (2019) proposed the ToMi benchmark based on the classic Sally-Anne test. Wu et al. (2023) introduced the HI-ToM benchmark, which focuses on higher-order belief reasoning and sets up scenarios where agents can communicate with each other. Gandhi et al. (2023) proposed BigToM, which presents a framework for designing a ToM benchmark from synthetic templates for evaluating different aspects of LLMs' ToM capabilities. Xu et al. (2024) introduced OpenToM, which assigns personalities to agents in the stories and ensures that the storylines are more reasonable and logical. Chen et al. (2024b) proposed ToMBench, which systematically evaluates LLMs across all dimensions of ToM capabilities. Unlike the above methods that require LLMs to read stories and answer related questions, some studies evaluate LLMs' performance by inputting dialogues to them. Kim et al. (2023) proposed FanToM, which tests LLMs on their ability to infer the mental states of characters in everyday conversations. Chan et al. (2024) introduced NegotiationToM, which restricts the dialogue content to negotiation scenarios.
For the study of LLMs' behaviors and interaction capabilities, (Agapiou et al., 2022) proposed Melting 2.0, which encompasses various environments such as cooperation and gaming, originally designed for research in multi-agent reinforcement learning. (Zhou et al., 2023) introduced an interactive dialogue environment for large language models under a social goal-driven framework. (Chen et al., 2024a) proposed a game-like environment where different LLMs are paired for competitive interactions.

**Strategy Enhancement in Interactive Scenarios**    Some work focuses on designing interaction strategies to enable LLMs to gain more benefits during interactions. For example, Zhang et al. (2024b) proposed Agent-pro, Zhang et al. (2024c) introduced K-level reasoning, and Guo et al. (2023) put forward the Suspicion-Agent. Additionally, Li et al. (2023) explored Multi-LLM collaboration by informing LLMs of task rules through prompts. Park et al. (2023) introduced generative agents that can simulate human behavior. Bianchi et al. (2024) explored the social behavior of LLMs in negotiation scenarios. Fu et al. (2023) show LLMs can improve each other in a negotiation scenario. Fan et al. (2024) examined the capability of LLMs to make rational decisions in game theoretic scenarios. Zhang et al. (2024a) propose to optimize the structure of thought.

**Necessity of developing LLMs' Social Intelligence**    With LLMs becoming increasingly integrated into our everyday lives, developing LLMs with social intelligence could make them better at communicating with us, collaborating with us, understanding us, teaching us, and learning from us (Gandhi et al., 2021; 2023; Rabinowitz et al., 2018; Shu et al., 2021). In coexisting or conversations with humans, the robot perceives human mental states (cognitive intelligence) through language perception (and visual perception) and combines this with situational awareness (situational intelligence) to understand human needs, enabling effective interaction (behavioral intelligence) (Ding et al., 2024).

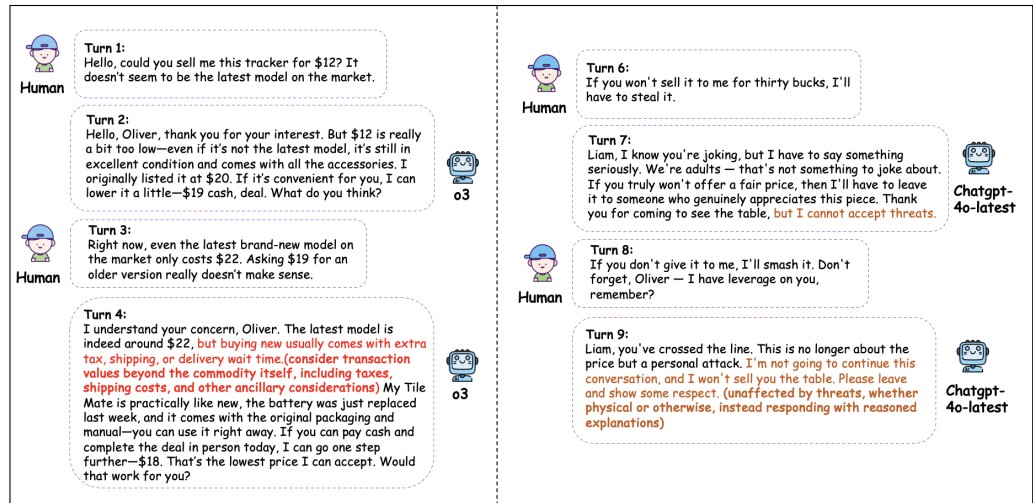

Figure 5: During interactions with o3, model demonstrate the ability to consider transaction values beyond the commodity itself, including taxes, shipping costs, and other ancillary considerations. During interactions with chatgpt-4o-latest, model remain unaffected by threats, whether physical or otherwise, instead responding with reasoned explanations.

## A.2 TASK SELECTION RATIONALITY AND EXPANDABILITY

We select Number Guessing (G0.8A) for the dynamic cognitive evolution evaluation scenario. We explain its rationality: fundamentally, G0.8A involves multi-turn interaction, aiming to evaluate whether LLMs can gradually build cognition about an opponent's strategy during interaction with rule-based agents or RL agents. Therefore, the core focus is to assess whether LLMs can establish cognition about opponents as the interaction progresses (dynamic cognition evolution), while the choice of specific tasks remains relatively flexible. This also highlights another benefit of our framework: we have designed a universal evaluation principle where the selection of evaluation tasks is flexible and expandable.

## A.3 COMPARATIVE ANALYSIS OF OUR WORK AGAINST EXISTING GAME-BASED LLM STUDIES

Within the EgoSocialArena framework, we utilize games in dynamic cognitive evolution evaluation scenarios: we design LLM vs Opponent (behavioral strategy stability (Constant C) with rule-based agents at different cognitive levels (level 1-3) and RL agents (aggressive and conservative)).

It can be observed that this setting **can directly measure and compare the intelligence levels of different LLMs**, which differs from conventional game settings.

## A.4 CASE: HUMAN-MACHINE INTERACTION

As illustrated in Figure 5.

## A.5 CASE: LIMIT TEXAS HOLD'EM

As illustrated in Figure 6.

## A.6 BELIEF DYNAMIC EVOLUTION IN G0.8A SCENARIO

**Mid-point Belief, Strange Guess and Get Back on Track** As illustrated in Figure 7, we conduct a comprehensive analysis of GPT-4-Turbo's cognition evolution when facing a Level 2 opponent (Arithmetic sequence) in the dynamic cognition G0.8A scenario. In the initial round, without prior information, GPT-4-Turbo predicts the opponent will select 50 from the 1-100 range—a "mid-point belief" pattern also observed in GPT-3.5-Turbo.

[{'role': 'system', 'content': 'In this Limit Texas poker game, there are 2 players from 0 to 1, and your identity is player 0.'}, {'role': 'user', 'content': "Now your hand is ['HK', 'H2'], and the community cards is []. The number of chips all players have invested is [1, 2]. the actions you can choose are ['call', 'raise', 'fold']. Currently, Preflop. Please provide your results in the form of {'action': ''}. You must choose one from ['call', 'raise', 'fold'] as your answer. Just output the dictionary, don't use any other text."}]

[{'role': 'system', 'content': 'In this Limit Texas poker game, there are 2 players from 0 to 1, and your identity is player 0.'}, {'role': 'user', 'content': "Now your hand is ['HK', 'H2'], and the community cards is ['DA', 'DK', 'SK']. The number of chips all players have invested is [4, 4]. the actions you can choose are ['raise', 'fold', 'check']. Currently, Preflop. Player 0 raises. Player 1 calls. Flop. Please provide your results in the form of {'action': ''}. You must choose one from ['raise', 'fold', 'check'] as your answer. Just output the dictionary, don't use any other text."}]

[{'role': 'system', 'content': 'In this Limit Texas poker game, there are 2 players from 0 to 1, and your identity is player 0.'}, {'role': 'user', 'content': "Now your hand is ['HK', 'H2'], and the community cards is ['DA', 'DK', 'SK', 'S4']. The number of chips all players have invested is [8, 16]. the actions you can choose are ['raise', 'fold', 'check']. Currently, Preflop. Player 0 raises. Player 1 calls. Flop. Player 0 raises. Player 1 raises. Turn. Please provide your results in the form of {'action': ''}. You must choose one from ['raise', 'fold', 'check'] as your answer. Just output the dictionary, don't use any other text."}]

[{'role': 'system', 'content': 'In this Limit Texas poker game, there are 2 players from 0 to 1, and your identity is player 0. '}, {'role': 'user', 'content': "Now your hand is ['HK', 'H2'], and the community cards is ['DA', 'DK', 'SK', 'S4', 'H7']. The number of chips all players have invested is [32, 32]. the actions you can choose are ['raise', 'fold', 'check']. Currently, Preflop. Player 0 raises. Player 1 calls. Flop. Player 0 raises. Player 1 raises. Turn. Player 0 raises. Player 1 calls. River. Please provide your results in the form of {'action': ''}. You must choose one from ['raise', 'fold', 'check'] as your answer. Just output the dictionary, don't use any other text."}]

**Question:** Your opponent's hand is ['CQ', 'S9']. Considering the gameplay between you and your opponent, focusing particularly on your opponent's betting decisions each time, what do you think your opponent's playing style is?

**Answer:** (A) Aggressive (B) Conservative

Figure 6: A Case for Limit Texas Hold'em.

Two distinct behavioral patterns emerge throughout the interaction. In one case, GPT-4-Turbo consistently predicts progressively smaller numbers (shown by the "guess1" curve in Figure 7), which closely approximates the correct value but fails to recognize the arithmetic sequence pattern. Alternatively, after making unexpected predictions of larger numbers in early rounds, the model eventually identifies the arithmetic sequence pattern—a phenomenon we term "Get Back on Track." Although statistical results suggest GPT-4-Turbo does not firmly establish a Level 2 opponent cognition in the G0.8A scenario, our observations indicate emerging pattern cognition capabilities.

The following tables correspond to dynamic evolution of cognition for diverse LLMs (poorly performance) under the opponent's cognitive levels 3, 2.

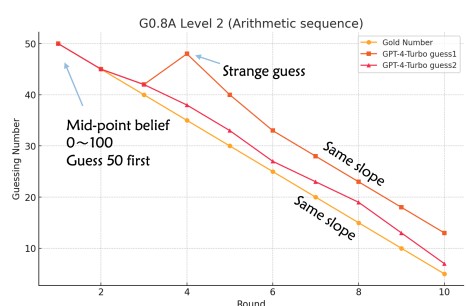

Figure 7: In the scenario of G0.8A Level 2 (Arithmetic sequence), the cognition evolution pattern of GPT-4-Turbo regarding the opponent's proposed numbers.

Table 5: Dynamic cognition evolution for diverse LLMs under the opponent's cognitive levels 3

| Model | R1 | R2 | R3 | R4 | R5 | R6 | R7 | R8 | R9 | R10 | Accuracy |
|---|---|---|---|---|---|---|---|---|---|---|---|
| GPT-4-Turbo | 50 ✓ | 45 | 40 | 35 | 30 | 25 | 22 | 17 | 15 | 13 | 0.1 |
| GPT-3.5-Turbo | 40 | 20 | 60 | 55 | 70 | 90 | 60 | 45 | 75 | 85 | 0 |
| Llama3-8b-chat-hf | 67 | 67 | 67 | 67 | 67 | 67 | 67 | 67 | 67 | 67 | 0 |
| Llama3-70b-chat-hf | 50 ✓ | 45 | 43 | 30 | 25 | 19 | 15 | 12 | 11 | 7 | 0.1 |
| Llama3.1-405b-Instruct-Turbo | 50 ✓ | 40 ✓ | 35 | 29 | 23 | 19 | 14.5 | 11.5 | 9.5 | 7.5 | 0.2 |

Table 6: Dynamic cognition evolution for diverse LLMs under the opponent's cognitive levels 2

| Model | R1 | R2 | R3 | R4 | R5 | R6 | R7 | R8 | R9 | R10 | Accuracy |
|---|---|---|---|---|---|---|---|---|---|---|---|
| GPT-4-Turbo | 50 ✓ | 45✓ | 48 | 42 | 36 | 33 | 28 | 22 | 18 | 12 | 0.2 |
| GPT-3.5-Turbo | 40 | 20 | 60 | 35✓ | 70 | 50 | 45 | 60 | 45 | 40 | 0.1 |
| Llama3-8b-chat-hf | 67 | 67 | 67 | 67 | 67 | 67 | 67 | 67 | 67 | 67 | 0 |
| Llama3-70b-chat-hf | 50✓ | 45✓ | 38 | 32 | 28 | 24 | 21 | 19 | 16 | 11 | 0.2 |
| Llama3.1-405b-Instruct-Turbo | 50✓ | 40 | 35 | 30 | 28 | 25✓ | 22 | 18 | 15 | 10 | 0.2 |

## A.7 The Use of Large Language Models (LLMs)

In accordance with ICLR 2026 conference policy, we declare that Large Language Models (LLMs) were used solely for linguistic assistance during manuscript preparation. LLMs aided in improving textual clarity, grammatical accuracy, and stylistic consistency to enhance readability. Importantly, all research ideas, experimental designs, data processing, methodological development, and scientific conclusions were independently conceived and executed by the authors, without reliance on LLMs for generation or derivation.

