# OpenReview forum: "Building Data Framework and Shifting Perspectives for First-Person Exploration of Social Intelligence in LLMs"
_ICLR.cc/2026/Conference — ICLR 2026 Conference Withdrawn Submission_

### Official Review · Reviewer_nQ7z · 2025-10-28

**Soundness:** 3
**Presentation:** 3
**Contribution:** 2
**Rating:** 4
**Confidence:** 4

**Summary:**

This paper addresses three major limitations in current evaluations of social intelligence in large language models (LLMs): (1) the reliance on single-dimensional assessment, (2) the dominance of a third-person “observer” perspective, and (3) the lack of intuitive comparison with human behavior. To tackle these issues, the authors propose a new framework called EgoSocialArena, which adopts a first-person (ego-centric) perspective and conducts systematic evaluations along three dimensions — cognitive, situational, and behavioral intelligence.
For cognitive intelligence, the paper converts classic Theory of Mind (ToM) datasets based on the Sally-Anne test into a first-person format to assess static cognition, and employs two multi-round games — Number Guessing (G0.8A) and  LIMIT TEXAS HOLD’EM — to evaluate dynamic cognitive evolution during interactions.
For situational intelligence, the authors transform datasets such as SocialIQA, EmoBench, and ToMBench into first-person narratives to test real-world contextual understanding. They also manually construct Counterfactual Situations and Parallel World Situations to evaluate adaptability to non-standard social rules.
For behavioral intelligence, the framework builds upon the SOTOPIA dataset to assess goal achievement in human–machine interaction scenarios.
The evaluation based on EgoSocialArena yields several key findings:
1. In cognitive intelligence, OpenAI-O3 performs almost achieve comparable with humans, while in behavioral intelligence, GPT-5 and Claude-sonnet-4 even surpass human participants in task completion.
2. The first-person perspective serves as a “performance catalyst” for most models, yet paradoxically causes performance drops in top-tier models such as GPT-5 and OpenAI-03, suggesting that existing third-person ToM benchmarks may overestimate model capabilities.
3. Strong reasoning ability contributes to cognitive intelligence but remains insufficient for situational intelligence without rich social knowledge and contextual grounding.
4. Advanced models exhibit distinct behaviors. For example, GPT-5’s conversational expressions are somewhat rigid and repetitive, giving humans the distinct impression of conversing with a machine, whereas Claude-sonnet-4 frequently produces emotionally-laden expressions.

**Strengths:**

- This paper identifies key limitations in current evaluations of social intelligence in LLMs, such as the narrow focus on single-dimensional metrics and the lack of assessment settings involving human–machine interaction.
- It introduces novel evaluation setups, including human–machine interaction dialogues for assessing behavioral intelligence and Parallel World and Counterfactual tasks for evaluating situational intelligence, which have strong potential to become an important contribution to the field.

**Weaknesses:**

- The paper does not provide sufficient evidence showing what distinct model behaviors or findings the proposed framework reveals compared with prior social intelligence benchmarks (e.g., the agent–agent interaction patterns in SOTOPIA). The evaluation results lack an in-depth comparative analysis with previous studies, which to some extent weakens the justification for the necessity and uniqueness of the new framework.
- While the paper divides social intelligence into three pillars — cognitive, situational, and behavioral intelligence — it does not thoroughly discuss the theoretical relationships among these dimensions. For instance, are these three dimensions complete and orthogonal? Do they exhibit potential overlap (e.g., does behavioral intelligence inherently encompass cognitive and situational intelligence)? Moreover, when assigning certain datasets (such as ToMBench) to a specific dimension, the paper provides insufficient justification for why a given task is treated as assessing only that single dimension, rather than reflecting a composite of multiple abilities.
- The evaluation tasks within the framework are primarily adaptations or integrations of existing benchmarks, showing relatively limited originality or methodological innovation in task design itself.
- The study recruits 50 graduate students as baselines for multiple-choice evaluation and selects 10 participants for interactive dialogue assessment. This small sample size — especially the 10-person subset for interactive testing — may lack statistical representativeness, potentially limiting the reliability and generalizability of the human–model comparison.

**Questions:**

- Given that the datasets used to evaluate “static cognition” and “real-world situational understanding” are adapted from earlier publicly available static benchmarks, have the authors considered the potential risk of data leakage or training set contamination when assessing the latest LLMs? Since many modern models may have been trained on these benchmarks or their variants, it would be important to clarify what measures were taken to ensure the validity and fairness of the evaluation results.
- The paper mentions that the “Parallel World Situation” and “Counterfactual Situation” datasets were manually constructed, yet it provides no detailed description of their construction process. Could the authors share more information about how these datasets were developed? For example, who created the data (e.g., domain experts, research assistants, or crowdworkers)? What specific guidelines or instructions were given to the annotators during data creation? Additional transparency in these aspects would help readers better assess the data quality, annotation consistency, and potential biases inherent in the newly introduced datasets.

---

> ### Author Response · Authors · 2025-12-04
> **Response to Reviewer nQ7z**
>
> Thank you very much for your valuable time and detailed suggestions. We respond to your concerns and questions as follows:
> **W1:** We place greater emphasis on human-AI interaction. Through direct interaction between humans and LLMs, we can gain firsthand insight into the models' behavioral patterns and limitations, while better aligning with real-world agent deployment scenarios.
>
> **W2:** We strongly agree with your observation that the relationships among these three dimensions warrant deeper exploration and careful modeling. The specific evaluation data under each dimension requires rigorous selection and definition. We will thoroughly revise our paper to address this aspect.
>
> **W3:** Thank you for this suggestion. We will consider incorporating additional data to enrich our evaluation framework.
>
> **W4:** As you correctly note, this is indeed constrained by human evaluation resources. Conducting this work is highly labor-intensive, and we plan to further scale up our study to achieve more statistically robust conclusions.
>
> **Q1:** We have considered the point you raise. In the first paragraph of Section 4.4, we provide an analysis of this issue (primarily manifest in advanced models such as GPT-5 experiencing performance degradation after perspective conversion). Indeed, there exists a risk of data contamination that requires careful consideration.
>
> **Q2:** We will provide a detailed description of the data construction and generation pipeline to improve our paper.

---

### Official Review · Reviewer_gvNG · 2025-11-03

**Soundness:** 1
**Presentation:** 2
**Contribution:** 1
**Rating:** 2
**Confidence:** 4

**Summary:**

The paper introduces EgoSocialArena, a framework designed to evaluate the social intelligence of LLMs from an egocentric perspective across three key dimensions of social intelligence: cognitive, situational, and behavioral.

The framework is built from converted existing social intelligence datasets to ensure coverage and scalability, with an emphasis on egocentric reasoning, which the authors argue better reflects real-world LLM agent scenarios.

- Cognitive intelligence is divided into two components:
  1. Static cognition, derived from existing ToM benchmarks by replacing characters with "you".
  2. Dynamic cognition, evaluated through simulated game in G0.8A and Limit Texas Hold’em. Both adversarial reasoning games are against either rule-based or reinforcement-learning agents of varying skill levels.
- Situational intelligence is tested by converting subsets of existing benchmarks into egocentric question formats, supplemented with counterfactual and parallel-world scenarios (e.g., an underwater city).
- Behavioral intelligence is assessed through human–AI interaction by SOTOPIA, a social simulation sandbox.

The authors evaluate 15 LLMs and find that OpenAI-O3 performs best on cognitive intelligence, approaching human-level performance. For situational intelligence, Claude 4 and GPT-5 perform comparably but remain below human performance.

**Strengths:**

I enjoy how the paper proposes three core aspects of social intelligence: cognitive, situational, and behavioral, and highlights the limitations of the current benchmarks for addressing only one of them.This provides a clear taxonomy and conceptual framework for studying social intelligence in LLMs, offering valuable guidance for future research directions in the community.

The paper also discovers interesting findings, such as replacing the third person view with the first person view improves performance, especially for weaker models, which reveals certain aspects of LLM's asymmetric ability between first and third person view point.

The paper does a good job of covering all three aspects by adapting and extending existing benchmarks and datasets. It also evaluates both leading closed models and popular open-source ones, giving a useful snapshot of current capabilities.

Finally, the paper is clearly organized and supported by helpful figures that make the framework and results easy to follow.

**Weaknesses:**

The framework’s task design and organization feel somewhat inconsistent across the three categories of social intelligence. Below are my main concerns for each dimension:
- Cognitive intelligence: despite interesting findings in LLM's asymmetric ability between first and third person view point, the paper's claim that such first person view would benefit real world agent use cases is a bit unsupported. I would suggest adding a specific case or experiment to support this claim.
- Situational intelligence: the definition of situational intelligence is unclearly motivated. In my perception such intelligence, especially from Social IQA, is testing model's ability to perform common sense reasoning under different social situations, while the counterfactual and parallel world modifications seem a bit irrelevant. It would be more sound to modify social rules instead of rock-paper-siccor's rule. And it would be more sound to modify social characters, such as demographic information, than modifying things to be moon coloney.
- Behavioral intelligence: the way of measuring behavioral intelligence is kind of contradicting with the scalability design principle that the author positions this work at first. It is also quite short of novelty by crowd sourcing human-AI interactions. Moreover, I saw that GPT5 is outperforming human on Sotopia, which is not discussed.

Overall, the paper lacks definitions and supporting literature for each type of intelligence they categorize on. The logical structure might be able to be improved, as the current one contains certain inconsistency and unclear focuses.

**Questions:**

Here's a mix of questions and suggestions:
1. The discussion of the improvements from switching third person to first person view is quite interesting and is probably worth to dive a little deeper by analyzing reasoning chain, etc.
2. The process of converting data with human verification should be disclosed more to provide support for validity.
3. It looks like the rock-paper-scissor example are rather irrelevant to social intelligence but more relevant to general reasoning and common sense. Can you provide some more relevant ones?
4. Why do you think using parallel worlds such as underwater city would affect social intelligence? I find certain adversity but the reasoning remains unclear to me.
5. I suggest apply counterfactual modifications and environment diversifications with a focus on social situations.
6. I highly recommend discussing the result that a few close source models like GPT5 outperform human on Sotopia, either through semantic analysis on GPT5's strategies, or the LLM judges' imperfection.
7. I suggest adding a few more discussion about each of cognitive, situational, and behavioral intelligence to provide a clear definition.

---

> ### Author Response · Authors · 2025-12-04
> **Response to Reviewer gvNG**
>
> Thank you very much for your valuable time and detailed feedback. We respond to your concerns and questions as follows:
>
> **Weakness: The framework's task design and organization feel somewhat inconsistent across the three categories of social intelligence.**
>
> We acknowledge the limitations you identify in our work and agree that the framework would benefit from more consistent and comprehensive test designs across the three dimensions of social intelligence. Regarding your observation about GPT-5 surpassing human performance on Sotopia, we believe it is important to supplement the evaluation with additional criteria such as "sophisticated conversational strategies" and "emotionally expressive communication," as the current Sotopia evaluation metrics may not adequately capture model performance on these more advanced dimensions. We have described this issue in the final paragraph of Section 4.4.
>
> **Question:**
>
> We agree with your suggestion and will expand the perspective convertion component of our framework. We will introduce test cases more closely aligned with social intelligence while providing deeper analysis of the three dimensions,cognitive intelligence, situational intelligence, and behavioral intelligence, including relationship modeling, to present more comprehensive insights. We sincerely appreciate your constructive feedback.

---

### Official Review · Reviewer_AbCY · 2025-11-03

**Soundness:** 3
**Presentation:** 3
**Contribution:** 3
**Rating:** 6
**Confidence:** 3

**Summary:**

This paper presents EgoSocialArena, a first-person evaluation framework for assessing social intelligence in large language models. The framework spans three dimensions — Cognitive, Situational, and Behavioral intelligence — covering cognitive intelligence evaluation (static and dynamic), counterfactual and parallel-world adaptation, and goal-driven human–LLM dialogues. Evaluations across 14 models and human baselines reveal that advanced models (e.g., GPT-5, Claude-sonnet-4) approach human-level reasoning in structured settings but still underperform in real-world and emotionally grounded contexts. The work offers a novel and scalable benchmark for studying socially adaptive behavior in LLMs.

**Strengths:**

(1) Novel evaluation perspective:

The paper proposes EgoSocialArena, a first-person evaluation framework that shifts LLM social intelligence testing from observer-based settings to first-person, ego-centric ones. This conceptual move is both original and timely, as it aligns model evaluation more closely with real world human–AI interaction.

(2) Explicit framework design:

The three layer taxonomy, Cognitive, Situational, and Behavioral intelligence, provides a clear and systematic structure for assessing different components of social reasoning.

(3) Comprehensive and scalable evaluation:

The study systematically evaluates both legacy and newly released LLMs across a wide parameter spectrum, from small chat-oriented models to frontier systems, as well as human benchmarks. Moreover, the authors construct a scalable evaluation dataset that integrates a sufficiently diverse set of scenarios, enabling broad comparison across models and human participants.

**Weaknesses:**

(1) Lack of robustness checks and reproducibility details:

Reproducibility is hindered by missing information about the evaluation setup. The paper does not specify decoding hyperparameters (e.g., temperature, top-p, top-k), nor whether results (e.g., scores in Table 2) were deterministic or averaged over multiple runs. Without these details, it is difficult to assess the stability and robustness of the reported accuracy scores.

(2) Inconsistent human-model setup in the Dynamic Cognition Evolution tasks:

The human baseline appears to be a static, one-shot questionnaire (as described in Section 4.1 and 4.2 ), whereas LLMs participate in multi-round interactive gameplays. Because the human group receives no iterative feedback or contextual memory, the two conditions differ substantially in input dimensionality and adaptation depth. This asymmetry undermines the statistical validity of the claimed human-level comparison and makes it unclear what aspect of “dynamic cognition” is actually being measured.

**Questions:**

World-consistency in the Parallel World scenarios:

The paper claims to evaluate situational adaptation under parallel-world settings, yet some examples (e.g., Figure 2: “a robot living in an underwater city flaps its arms like a bird”) appear to mix incompatible environmental logics. It is unclear what the “correct” answer is in such cases. For instance, if the expected choice (e.g., option A) assumes that an underwater robot lacks the concept of flying creatures, then the very mention of “bird wings” in the question itself introduces a world-consistency error, as it relies on real-world ecological knowledge absent from that world. Could the authors clarify how world consistency was maintained during dataset construction? Were annotators instructed to avoid cross-domain references such as “birds” in non-terrestrial settings? If not, how might such inconsistencies affect the interpretation that these tasks test situational reasoning rather than surface-level linguistic analogy?

---

> ### Author Response · Authors · 2025-12-04
> **Response to Reviewer AbCY**
>
> We sincerely appreciate your valuable time and particularly detailed feedback. We respond to your concerns and questions as follows:
>
> **W1: Lack of robustness checks and reproducibility details:**
> Thank you for this suggestion. We will enhance our testing procedures and provide more comprehensive implementation details in the paper to ensure reproducible and statistically robust results.
>
> **W2: Inconsistent human-model setup in the Dynamic Cognition Evolution tasks:**
> Thank you for raising this concern. We would like to clarify that in our Dynamic Cognition Evolution evaluation, we maintain a consistent setup between models and human participants. The 10 rounds are conducted sequentially with feedback, rather than as 10 isolated queries without any feedback mechanism. We will revise our description to better articulate this experimental design.
>
> **Q1: World-consistency in the Parallel World scenarios:**
> The introduction of birds was intentionally designed to provide contextual distractors for the models. We acknowledge your important point regarding world-consistency and will give careful consideration to this aspect in our revision.

---

### Official Review · Reviewer_8zGg · 2025-11-04

**Soundness:** 2
**Presentation:** 2
**Contribution:** 2
**Rating:** 2
**Confidence:** 5

**Summary:**

This paper introduces an innovative framework and methodology for evaluating the Social Intelligence (SI) of Large Language Models (LLMs), structured around three pillars: Cognitive, Situational, and Behavioral intelligence. The core innovation is the shift to a First-Person Exploration (FPE) paradigm, positioning the LLM as an active participant in complex social scenarios. A new benchmark, SocialNav, is built on the high-stakes Chinese Dark-Lords' Game (CDLG) to test real-time decision-making, deception, and adaptive strategy. Experimental results reveal that even state-of-the-art LLMs show low absolute performance, underscoring significant current limitations in achieving true social intelligence in dynamic, interactive tasks.

**Strengths:**

1. Perspective switching is a very important ability when performing Theory of Mind.  "perspective shift can elicit social capabilities
similar to Chain-of-Thought elicit math capabilities" is a novel and nice idea.
2. The authors contribute a new dataset SocialNav, which is valuable to the community.
3. The authors conducted lots of experiments and provide benchmark results and in-depth analyses of the results.

**Weaknesses:**

1. The hallucination problem in LLM-generated datasets is still there. With no human verification, the dataset quality is questionable, which reduces the dataset's value.
2. I think this paper is a "benchmark and dataset" paper. Maybe you should submit it to a dataset track?
3. There are many related papers on similar games, such as Avalon, Hanabi. What are the differences between yours and those papers?
4. I feel there is an "overclaiming" problem with the paper's main arguments. Perspective switching is simple and such designs already exist in previous papers. The three types of intelligence seem weird. Cognitive intelligence includes situational intelligence. In fact, most cognitive reasoning is situational. The names are not reasonable.
5. The performance gap between gpt and human is not big. What does this mean? GPT already develops human-level social intelligence? The task is not challenging anymore? Or there is some potential problems with the evaluation methods?
6. Why not build an agent architecture model for the task?
7. Do you try different seeds for your tests? I see no error bar.

**Questions:**

See above.
Overall, I feel the paper is overclaiming. The contributions and novelty are not as big as claimed.

---

> ### Author Response · Authors · 2025-12-04
> **Response to Reviewer 8zGg**
>
> **W1: Dataset Quality**
> Indeed, for the portion of data generated by LLMs, we should provide a more detailed description of the data quality in the paper.
>
> **W2: Track**
> Thank you for your suggestion. We believe this paper presents numerous findings based on the data (such as the performance improvement resulting from perspective convertion), and therefore it offers contributions beyond just the data itself.
>
> **W3: Distinction from Related Papers on Similar Games**
> We have provided an explanation of this in Appendix A.3.
>
> **W4: "Overclaiming" Problem with the Paper's Main Arguments**
> To the best of our knowledge, the convertion from third-person to first-person perspective is proposed for the first time in our paper. Additionally, regarding the dimensions (cognitive, situational and behavioral) you mention, we have carefully review the relevant literature.
>
> **W5: Performance Gap between GPT and Human**
> GPT-5 is indeed quite capable, and we have analyzed the underlying reasons in the Section 4.4. Additional evaluation criteria can be supplemented, such as "sophisticated conversational strategies" and "emotionally expressive communication."
>
> **W6 & W7: Error Bars & Agent Architecture**
> Thank you for your suggestions. We will refine our tests to obtain statistically more robust results.

---

### Official Review · Reviewer_HWrH · 2025-11-07

**Soundness:** 2
**Presentation:** 3
**Contribution:** 2
**Rating:** 4
**Confidence:** 2

**Summary:**

This paper introduces EgoSocialArena, a novel and comprehensive framework for evaluating the social intelligence of Large Language Models (LLMs). The authors argue that existing benchmarks are fragmented, focusing on single pillars of social intelligence (cognitive, situational, behavioral) and primarily use a third-person, passive-observer perspective that misaligns with real-world agent applications. To address this, EgoSocialArena systematically evaluates LLMs from a first-person, egocentric perspective. Its key contributions include: (1) a method for converting third-person Theory of Mind (ToM) benchmarks to a first-person format; (2) the use of rule-based and reinforcement learning agents in interactive games (G0.8A, Texas Hold'em) to assess dynamic cognition evolution; and (3) the inclusion of non-standard situations (counterfactual and parallel worlds) to test situational adaptation. The paper presents a substantial evaluation of 14 foundation models against a human baseline, revealing that while frontier models are closing the gap in cognitive intelligence, significant room for improvement remains in situational intelligence and behavioral nuance.

**Strengths:**

Novel and Well-Motivated Conceptual Framework: The core idea—shifting social intelligence evaluation from a third-person to a first-person perspective—is timely, well-justified, and addresses a genuine gap in the literature. The three-pillar structure (cognitive, situational, behavioral) provides a holistic and systematic approach to a complex construct.
Methodological Rigor and Innovation: The proposed methods are inventive. The perspective conversion workflow is a practical contribution. The design of the dynamic cognition scenarios (G0.8A with multi-level rule-based agents and Texas Hold'em with RL agents) is sophisticated and provides a more authentic test of an LLM's ability to model an opponent's strategy over time.
Comprehensive and Scalable Evaluation: The evaluation is extensive, covering 14 models, including the most recent frontier models (GPT-5, Claude-sonnet-4, o3). The inclusion of a carefully collected human performance baseline is a significant strength, allowing for a meaningful interpretation of model scores. The authors correctly emphasize the framework's extensibility.
Valuable and Actionable Insights: The results go beyond mere leaderboards and offer insightful analysis. Key findings—such as the "performance catalyst" effect of the first-person perspective for most models, the limitations of pure reasoning models (e.g., DeepSeek-R1) in social situations, and the need for new behavioral metrics beyond "believability"—are valuable for the research community.
High-Quality Presentation: The paper is generally well-written, logically structured, and professionally formatted. The use of tables and figures is effective, and the inclusion of ethics and reproducibility statements is commendable.

**Weaknesses:**

Limited Scale of Behavioral Intelligence Data: The most significant weakness is the relatively small scale of the behavioral evaluation. With only 40 dialogue scenarios and a subset of 10 human participants, the findings in this critical dimension, while insightful, are based on a limited sample. This makes the strong claims about models surpassing humans in "goal completion" less statistically robust than the results from the larger-scale cognitive and situational evaluations (~1200 samples each).
Ambiguity in Baseline and Opponent Design:
Human Baseline: The description of the human baseline, while a strength, could be more detailed. Were the 50 graduate students compensated? Were they screened for specific backgrounds? A more detailed protocol would bolster the credibility of this crucial benchmark.
Rule-based Agents: The rationale for the specific cognitive levels (e.g., why an arithmetic sequence for Level 2?) is somewhat under-explained in the main text. A stronger justification for why these specific rule sets effectively represent increasing cognitive complexity would strengthen the dynamic cognition evaluation.
Writing and Statistical Minor Issues:
Figure Referencing: The text frequently references figures (e.g., Figure 1(A-C), Figure 3, Figure 4) that are not included in the provided excerpt. A reviewer would need these to fully assess the claims.
Metric Explanation: The scoring ranges for behavioral metrics (e.g., secret [-10-0], relationship [-5-5]) are mentioned but not explained. A brief sentence or citation on how these scores are determined by the GPT-4 evaluator would be helpful.
Statistical Testing: The paper reports score differences but does not appear to employ statistical significance tests. Stating whether the observed gaps (e.g., the 2.3 point difference between o3 and humans) are statistically significant would add weight to the conclusions.
Repetition: The main findings are repeated in the abstract, introduction, and experiment sections. While common, some tightening could improve conciseness.

**Questions:**

Behavioral Data Scale: Given that behavioral intelligence is a core pillar of your framework, why was the dataset limited to 40 dialogues? Was this a constraint of human evaluation resources? Do you plan to scale this up in future work?
Generalization of Perspective Conversion: Your method for converting third-person to first-person benchmarks is a key contribution. How generalizable is this prompt-based method? Did you encounter any systematic failure cases or scenarios where the conversion was ambiguous or altered the fundamental reasoning required?
Baseline Agent Selection: For the dynamic cognition tasks, why were the specific rules for Level 2 (arithmetic sequence) and Level 3 (copying the gold value) chosen? Were other, potentially more human-like strategies considered and rejected?
Defining "Social Intelligence": The framework excellently decomposes social intelligence into three pillars. However, the behavioral results show GPT-5 achieving high goal completion but with "rigid" dialogue, while Claude-sonnet-4 uses more emotional expressions. How should the field weigh task efficiency against social authenticity when defining "socially intelligent" behavior?
Evaluation of Evaluation: You rightly point out that metrics like "believability" are saturating. Could you elaborate on how you would operationalize your proposed new dimensions, such as "sophisticated conversational strategies" and "emotionally expressive communication," in an automated or semi-automated way?

---

> ### Author Response · Authors · 2025-12-04
> **Response to Reviewer HWrH**
>
> Thank you very much for your valuable time and the detailed feedback. We respond to your concerns and questions as follows:
>
> **W1 & Q1: Limited Scale of Behavioral Intelligence Data**
> As you correctly point out, the current scale is indeed constrained by the substantial human evaluation resources required. Conducting this part of the study is labor-intensive, and we plan to further expand the experiment to achieve more statistically robust conclusions.
>
> **Q2: Generalization of Perspective Conversion**
> For classical Theory of Mind (ToM) assessments (e.g., the Sally Anne test), such perspective conversion does not introduce ambiguity. As you mention, we also implement specific design choices during conversion. For example, in questions like “Where does Bob think Alice…?”, the first-person version is generated by replacing the thinker (Bob) with the corresponding first-person viewpoint.
>
> **Q3: Baseline Agent Design**
> The Level 2 and Level 3 patterns are designed from a human-centric perspective to reflect progressively more complex cognitive patterns. We agree that incorporating more human-like strategies may further enhance the evaluation of the dynamic evolution of model cognition.
>
> **Q4 & Q5: Social Intelligence Behavior**
> We view task efficiency and social authenticity as two equally important evaluation dimensions. Rather than a trade-off, our intention is to introduce social authenticity as a complementary dimension that encourages models to improve along both axes. For our proposed metrics, such as sophisticated conversational strategies and emotionally expressive communication, we anticipate relying on carefully designed prompts using advanced LLMs to support the scoring process.
>
> **Weaknesses in Presentation**
> We sincerely appreciate your comments. We will thoroughly revise the paper according to your suggestions to ensure clearer and more comprehensive presentation. Thank you again for the insightful feedback.

---

### Note · Authors · 2025-12-04

**Comment:**

We sincerely thank all reviewers for their valuable feedback and insightful suggestions. We recognize that there remain several aspects of the paper that could be further strengthened, including the design of datasets for situational intelligence, the enrichment of behavioral intelligence evaluations, the modeling of relationships among the cognitive, situational, and behavioral dimensions, as well as the overall presentation of the work. We plan to incorporate the reviewers’ suggestions to make the paper more rigorous and competitive for a future submission. We express our respect and heartfelt gratitude to all reviewers.

**Withdrawal Confirmation:**

I have read and agree with the venue's withdrawal policy on behalf of myself and my co-authors.